# Mesenchymal Stem/Stromal Cells Derived from Cervical Cancer Promote M2 Macrophage Polarization

**DOI:** 10.3390/cells12071047

**Published:** 2023-03-30

**Authors:** Víctor Adrián Cortés-Morales, Luis Chávez-Sánchez, Leticia Rocha-Zavaleta, Sandra Espíndola-Garibay, Alberto Monroy-García, Marta Elena Castro-Manrreza, Guadalupe Rosario Fajardo-Orduña, Teresa Apresa-García, Marcos Gutiérrez-de la Barrera, Héctor Mayani, Juan José Montesinos

**Affiliations:** 1Mesenchymal Stem Cells Laboratory, Oncology Research Unit, Oncology Hospital, National Medical Center (IMSS), Mexico City 06720, Mexico; 2Programa de Doctorado en Ciencias Biomédicas, Universidad Nacional Autónoma de Mexico (UNAM), Mexico City 04510, Mexico; 3Immunology Medical Research Unit, Pediatric Hospital, National Medical Center (IMSS), Mexico City 06720, Mexico; 4Departamento de Biología Molecular y Biotecnología, Instituto de Investigaciones Biomédicas, Universidad Nacional Autónoma de Mexico, Ciudad de Mexico 04510, Mexico; 5Immunology and Cancer Laboratory, Oncology Research Unit, Oncology Hospital, National Medical Center (IMSS), Mexico City 06720, Mexico; 6Immunology and Stem Cells Laboratory, Multidisciplinary Unit of Experimental Research Zaragoza, FES, Zaragoza, National Autonomous University of Mexico, Mexico City 09230, Mexico; 7Oncology Research Unit, Oncology Hospital, National Medical Center (IMSS), Mexico City 06720, Mexico; 8Hematopoietic Stem Cells Laboratory, Oncology Research Unit, Oncology Hospital, National Medical Center (IMSS), Mexico City 06720, Mexico

**Keywords:** tumor-derived mesenchymal stem/stromal cells, cervical cancer, macrophage polarization, immunoregulation

## Abstract

Macrophages with the M2 phenotype promote tumor development through the immunosuppression of antitumor immunity. We previously demonstrated the presence of mesenchymal stem/stromal cells (MSCs) in cervical cancer (CeCa-MSCs), suggesting an immune protective capacity in tumors, but to date, their effect in modulating macrophage polarization remains unknown. In this study, we compared the capacities of MSCs from normal cervix (NCx) and CeCa to promote M2 macrophage polarization in a coculture system. Our results demonstrated that CeCa-MSCs, in contrast to NCx-MSCs, significantly decreased M1 macrophage cell surface marker expression (HLA-DR, CD80, CD86) and increased M2 macrophage expression (CD14, CD163, CD206, Arg1) in cytokine-induced CD14^+^ monocytes toward M1- or M2-polarized macrophages. Interestingly, compared with NCx-MSCs, in M2 macrophages generated from CeCa-MSC cocultures, we observed an increase in the percentage of phagocytic cells, in the intracellular production of IL-10 and IDO, the capacity to decrease T cell proliferation and for the generation of CD4^+^CD25^+^FoxP3^+^ Tregs. Importantly, this capacity to promote M2 macrophage polarization was correlated with the intracellular expression of macrophage colony-stimulating factor (M-CSF) and upregulation of IL-10 in CeCa-MSCs. Furthermore, the presence of M2 macrophages was correlated with the increased production of IL-10 and IL-1RA anti-inflammatory molecules. Our in vitro results indicate that CeCa-MSCs, in contrast to NCx-MSCs, display an increased M2-macrophage polarization potential and suggest a role of CeCa-MSCs in antitumor immunity.

## 1. Introduction

Mesenchymal stem/stromal cells (MSCs) are a heterogeneous cell population present in the stroma of different tissues that have the ability to differentiate into different cell lines, such as adipocytes, chondrocytes, osteocytes and neurons [1]. MSCs have the ability to migrate to sites of inflammation and have been isolated from different types of cancer, such as osteosarcoma [2], gastric cancer [3], neuroblastoma [4] and ovarian cancer [5]. These cells have been described as constituents of the tumor microenvironment (TME), where it has been shown that they have the ability to regulate tumor growth and progression through immunological mechanisms that have not been fully elucidated [6,7,8]. Our research group reported the presence of MSCs in both normal cervix (NCx) and cervical cancer (CeCa). We found that CeCa-MSCs, compared to NCx-MSCs, decrease the expression of HLA class I molecules in tumor cells through the secretion of interleukin-10 (IL-10), which causes cytotoxic T lymphocytes to fail to recognize neoplastic cells, thus suggesting that tumor-derived MSCs provide immune protection to tumor cells by inducing the downregulation of HLA class I molecules [9].

MSCs have been shown to possess an immunoregulatory capacity, as they can interact with cells of the innate and adaptive immune systems and decrease their inflammatory phenotype through the production of immunosuppressive molecules such as interleukin-4 (IL-4), IL-10, transforming growth factor beta (TGF-β), prostaglandin E2 (PGE2) and indoleamine 2-3 dioxygenase (IDO), among others [10]. They also have the ability to recruit immunosuppressive cells such as myeloid-derived suppressor cells [11] and regulatory T cells (Tregs) [12]. Some of these mechanisms have been shown to contribute to inhibiting the immune response against tumors in lung [13], pancreatic [14] and gastric [15] cancer. We have reported the immunoregulatory capacity of MSCs in CeCa; these cells suppress the response of T lymphocytes through the purinergic pathway, in which two membrane ectoenzymes of these cells participate: CD39, which hydrolyzes adenosine triphosphate/adenosine diphosphate (ATP/ADP) nucleotides to generate the respective nucleotides, which are hydrolyzed by CD73, which converts adenosine monophosphate (AMP) into adenosine, an inhibitory metabolite of the T lymphocyte response [16], an effect that potentiates immune protection in tumor cells [17].

Macrophages are a type of immune cell that is involved in various stages of inflammatory processes. These cells have the ability to be polarized toward proinflammatory (M1) or anti-inflammatory (M2) phenotypes [18]. Previous studies indicate that bone marrow MSCs (BM-MSCs) and placenta, when in contact with macrophages, inhibit M1 polarization, favoring the switch to the M2 phenotype because they decrease the expression of the costimulatory molecules CD40, CD80 and CD86 as well as the secretion of the proinflammatory molecules tumor necrosis factor alpha (TNFα) and interleukin-1 beta (IL-1β), increase the expression of CD36, CD206 and arginase 1 (Arg1) phagocytic activity, and the secretion of IL-10 [19,20,21]. Furthermore, BM-MSCs generate macrophages that decrease the formation of Th1 and Th17 lymphocytes [22] and promote the generation of Tregs [21]. Macrophages with the M2 phenotype are present in the tumor microenvironment in different types of cancer and are identified as tumor-associated macrophages (TAMs), which support tumor growth through different functions, thus promoting chronic inflammation, immune suppression, angiogenesis and invasion/metastasis [23]. In this context, it is important to know if the MSCs present in CeCa have the immunosuppressive potential to promote M2 macrophage polarization and play a role in antitumor immunity; however, to date, this aspect is unknown.

CeCa patients with fewer M2 macrophages show a better response to chemotherapy after surgery and better survival [24]. The results of a previous study reported poor survival for patients with CeCa who presented TAMs that expressed PD-L1, a marker associated with M2 polarization [25]. Based on this background, in this work, we analyzed the ability of CeCa-MSCs to promote M2 macrophage polarization of CD14^+^ monocytes in in vitro cultures in the presence of cytokines that promote M1 or M2 polarization. For this purpose, we conducted a comparative study of MSCs derived from NCx and CeCa and their potential to decrease macrophage expression of M1 surface markers and increase secreted anti-inflammatory molecules. Similarly, we evaluated the potential of MSCs from both sources to increase both the expression of characteristic markers of an M2 population and their capacity for phagocytosis, generating Tregs, to show the presence of an M2 macrophage-polarized population. Finally, we analyzed the potential of MSCs to produce intracellular cytokines that have been reported to be involved in the M2 polarization of macrophages. To our knowledge, this is the first in vitro study comparing the macrophage polarization potential of MSCs derived from NCx and CeCa to determine the possible immunoregulatory role of CeCa-MSCs in the tumoral microenvironment.

## 2. Materials and Methods

### 2.1. Collection and Culture of MSCs

Samples of BM-MSCs (*n* = 6) were obtained as previously described [26]. The samples of NCx (*n* = 6) and CeCa (*n* = 6) MSCs were processed and cultured as previously reported [9]. MSCs at passages 4 and 5 were used in the experiments.

BM cells were collected according to institutional ethics guidelines, with informed consent from hematologically normal donors at Bernardo Sepulveda Hospital, National Medical Center, Mexican Institute for Social Security (IMSS), Mexico City, Mexico. We enriched the BM-MSC population using the negative selection procedure RosetteSep™ Human Mesenchymal Stem Cell Enrichment Cocktail system (StemCell Technologies, Vancouver, BC, Canada) and following the supplier’s instructions. The cells were resuspended in low-glucose Dulbecco’s modified Eagle medium (Lg-DMEM; Gibco BRL, Rockville, MD, USA) supplemented with 10% fetal bovine serum (FBS; Gibco BRL) and seeded at a density of 200,000 cells/cm^2^ into T25 cell culture flasks (Corning, Corning, NY, USA). Every 5 days, a medium change was performed. When the cultures reached 80% confluence, they were trypsinized (0.05% trypsin, 0.53 mM EDTA; Gibco BRL) and subcultured at a density of 10,000 cells/cm^2^ into T75 flasks (Corning). From the third passage the cells were cultured in 100 mm TC (tissue culture) dish at a density of 10,000 cells/cm^2^.

NCx and CeCa samples were collected according to institutional ethics guidelines with informed consent from normal donors at Troncoso Hospital and from cancer patients at Oncology Hospital, Mexican Institute for Social Security (IMSS), Mexico City, Mexico. NCx samples were obtained from hysterectomy surgery of normal subjects. CeCa samples were obtained from biopsies from patients in stage IIB and IIIB. CeCa biopsies were confirmed by the Pathology Department for diagnosis. Cervical biopsy samples were dissected into small pieces and washed with PBS 1× (Gibco BRL), after which they were digested with trypsin-EDTA (0.5%/0.2%) (Gibco BRL) for 20 min at 37 °C under constant stirring. Single-cell suspension was collected by flushing the tissue parts through a 100 mm nylon filter (Corning) and centrifuged to obtain the cell pellet. Total numbers of mononucleated and viable cells were determined, seeded and manipulated as described for BM. 

### 2.2. Characterization of Mesenchymal Stem/Stromal Cells

#### 2.2.1. Immunophenotype

The immunophenotype characterization of the MSCs was conducted using a previously described protocol [1]. Conjugated monoclonal antibodies against PE-Cy™5 mouse anti-human CD90, PE-Cy™7 mouse anti-human CD73, APC mouse anti-human CD13, FITC mouse anti-human HLA-ABC, APC mouse anti-human CD34, FITC mouse anti-human CD31, PE mouse anti-human CD45, PE mouse anti-human CD14 (BD Biosciences, San Diego, CA, USA), eFluor™450 mouse anti-human CD105, PE mouse anti-human CD29 (eBioscience, San Diego, CA, USA), PE/Cyanine7 mouse anti-human HLA-DR and PE/Cyanine7 mouse anti-human CD10 (BioLegend, San Diego, CA, USA) were used.

#### 2.2.2. Morphological Analysis

To evaluate their morphology, 8.75 × 10^3^ cells/cm^2^ were seeded in 35 mm TC dishes (Corning). When 70% confluent, the cells were stained with Wright’s stain (Sigma-Aldrich, St. Louis, MO, USA) and evaluated using a phase contrast microscope (Zeiss, Oberkochen, Germany).

#### 2.2.3. Differentiation Capacity

The evaluation of the adipogenic and osteogenic differentiation capacity of MSCs was performed using a previously reported protocol [9]. Adipogenic differentiation was verified by the visualization of lipid vacuoles stained with oil red O (Sigma-Aldrich). For osteogenic differentiation, alkaline phosphatase activity was determined using SIGMA FAST™BCIP/NBT (5-bromo-4-chloro-3-indolylphosphate/nitro blue tetrazolium) (Sigma-Aldrich). For chondrogenic differentiation, 7 × 10^5^ cells were seeded in 35 mm TC dishes (Corning), and when the cells reached a confluence of 70%, chondrogenic medium (Cambrex Bio Science, East Rutherford, NJ, USA) supplemented with 10 ng/mL transforming growth factor beta 3 (TGF-β3) (Peprotech, Cranbury, NJ, USA) was added; the culture continued for 21 days, changing the medium two times a week. Finally, the presence of cellular matrix in the cell monolayer was evaluated with Alcian Blue dye (Sigma-Aldrich).

#### 2.2.4. Evaluation of MSCs Proliferation Capacity

A total of 100,000 MSCs at passage 4 or 5 were cultured in 100 mm TC dishes, under the conditions indicated in Section 2.1. After 7 days of culture, MSCs were obtained with trypsin (GIBCO BRL) and the cell number was counted with trypan blue dye. The number obtained at the end of the assay was divided by the number of initial cells, obtaining the fold-change.

### 2.3. Obtaining and Culturing CD14^+^ Monocytes

CD14^+^ monocytes were obtained from peripheral blood mononuclear cells (PBMCs) from healthy adult donors. PBMCs were obtained using a density gradient with Lymphoprep at a density of 1.077 + 0.001 g/mL (STEMCELL Technologies). CD14^+^ cells were isolated from PBMCs by negative magnetic selection using Micro Beads and MACS LS columns (Miltenyi Biotec, Bergisch Gladbach, North Rhine-Westphalia, Germany) following the protocol provided by the supplier. CD14^+^ cells were incubated in RPMI-1640 medium (HyClone, Logan, UT, USA) supplemented with 10% fetal bovine serum (FBS) (Biowest, Nuaillé, PC, France), 2 mM L-glutamine (Biowest), 1× penicillin/streptomycin (Biowest) and 100 μg/mL gentamicin (Biowest) until use.

### 2.4. Macrophage Polarization

For M1 polarization, CD14^+^ monocytes were cultured in 45% RPMI-1640 medium (HyClone), 45% low-glucose DMEM (HyClone) and 10% FBS (Biowest) with inducer medium 1 (M1), which contained granulocyte-macrophage colony-stimulating factor (GM-CSF) (50 ng/mL) (Miltenyi Biotec), for 96 h. Then, the medium was removed, and lipopolysaccharides (LPS) (100 ng/mL) (Miltenyi Biotec) and interferon gamma (IFNγ) (50 ng/mL) (Peprotech) were added, and the cells were incubated for 48 h. To induce M2 polarization, CD14^+^ monocytes were cultured in 45% RPMI-1640 medium (HyClone), 45% low-glucose DMEM (HyClone) and 10% FBS (Biowest) with inducer medium 2 (M2), which contained M-CSF (50 ng/mL) (Peprotech), for 96 h. Then, the medium was removed, and new medium consisting of interleukin-4 (IL-4) (Peprotech) (50 ng/mL) and interleukin 13 (IL-13) (40 ng/mL) (Peprotech) was added, and the cells were incubated for 48 h. CD14^+^ monocytes without cytokines were cultured as a negative control, absence of inducer medium (M0).

### 2.5. Macrophage/MSC Cocultures

CD14^+^ monocytes were cocultured for 6 days in the presence or absence of BM, NCx or CeCa-MSCs at a ratio of 5:1 (CD14^+^: MSCs) in a system with cellular contact or a Transwell system with a pore size of 0.4 μm (Corning). The cocultures were maintained in the absence (control, M0) or presence of M1 or M2.

### 2.6. Phenotyping of Macrophage Membrane Markers

Macrophage membrane markers were evaluated by flow cytometry 6 days after initiating the previously described treatments. The macrophages were washed with 1× phosphate buffered saline (PBS) (Biowest), stained with 25 μL of the viability marker Ghost Dye Red 780 (TONBO biosciences, San Diego, CA, USA; 1 μL of the stock diluted in 1499 μL of PBS) for 15 min at room temperature, washed with 1× PBS and blocked with FBS (Biowest) at 4 °C for 10 min. After blocking with FBS, diluted antibodies were added to detect characteristic markers of the M1 macrophage population: PE mouse anti-human CD86 (eBioscience), BV421 mouse anti-human CD80 (BioLegend) and PE/Cyanine7 anti-human HLA-DR (BioLegend). A panel of antibodies was also used to detect characteristic markers of the M2 macrophage population: BV510 mouse anti-human CD14 (BioLegend), APC mouse anti-human CD163 (BioLegend) and PE-Cy™7 mouse anti-human CD206 (BD Biosciences). After adding the antibodies, the cells were incubated at 4 °C for 20 min. The cells were then fixed with 1% paraformaldehyde for 10 min at 4 °C and washed with PBS. Acquisitions were performed on a BD FACSCanto II flow cytometer (BD Biosciences) and analyzed with FlowJo (Ashland, OR, USA, V10 software) Macrophages grown in the absence of MSCs and under different polarization conditions were used as controls. The calculation of M1 and M2 polarization was obtained by individual experiments with MSCs derived from BM (*n* = 6), NCx (*n* = 6) and CeCa (*n* = 6), which were performed to determine the fold increase (compared to the control) in the intensity of positive expression of membrane molecules characteristic of M1 or M2 polarization in macrophages cocultured in cell contact for 6 days in vitro.

### 2.7. Phagocytosis Assay

Macrophages generated in the presence of MSCs without contact with a 0.4 µm Transwell (Corning), where the MSCs were seeded on the Transwell, and macrophages in the bottom well of a Transwell after 6 days of culture in the presence of cytokines that favor M0, M1 or M2 polarization were challenged for 45 min with *Escherichia coli* bioparticles from the pH Rodo kit (Invitrogen, Waltham, MA, USA) and treated as indicated following the supplier’s instructions. The cells were then fixed with 1% paraformaldehyde for 10 min at 4 °C, washed with PBS, and evaluated by flow cytometry. The acquisitions were performed on a BD FACSCanto II flow cytometer (BD Biosciences) and analyzed with FlowJo (Ashland). Macrophages grown in the absence of MSCs and under different polarization conditions were used as controls.

### 2.8. Evaluation of Intracellular Molecules

After 6 days of interaction between MSCs and macrophages, the cocultures were treated with the Golgi Stop reagent (BD Biosciences) to inhibit protein transport so as to evaluate intracellular molecules. After 5 h of treatment, the cocultures were washed with 1× PBS, stained with 25 μL of the viability marker Ghost Dye Red 780 (TONBO biosciences; 1 μL of the stock was diluted in 1499 μL of PBS) for 15 min at room temperature, washed with 1× PBS (Biowest) and blocked with FBS (Biowest) at 4 °C for 15 min. After blocking, APC mouse anti-human CD45 (BD Biosciences) was added to the cells, followed by incubation at 4 °C for 20 min and a wash with 1× PBS (Biowest). The cell membrane was permeabilized following the instructions of the FoxP3 staining buffer set kit (Invitrogen), and antibodies were added to the cells to evaluate macrophages: PE/Cyanine7 mouse anti-human IFNγ, PE mouse anti-human Arg1, BV421 mouse anti-human IL-10 (BioLegend) and APC mouse anti-human IDO (RyD Systems, Minneapolis, MA, USA). To evaluate MSCs, BV421 mouse anti-human IL-10 (BioLegend) and PE mouse anti-human M-CSF (RyD Systems) were added to the cells. The cells were subsequently washed with 1× PBS (Biowest). Acquisitions were performed on a BD FACSCanto II flow cytometer (BD Biosciences) and for Arg1 was performed on a spectral flow cytometer Aurora (Cytek Biosciences, Fremont, CA, USA) and analyzed with FlowJo V10 software. Macrophages grown in the absence of MSCs and under different polarization conditions were used as controls. In addition to the detection of membrane molecules, we included in the calculation of M1 and M2 polarization the expression of intracellular molecules (M1: IFNγ and M2: Arg1, IL-10 and IDO) which was analyzed by individual experiments with MSCs derived from BM (*n* = 6), NCx (*n* = 6) and CeCa (*n* = 6), which were performed to determine the fold increase (compared to the control) in the intensity of positive expression of intracellular molecules characteristic of M1 or M2 polarization in macrophages cocultured in cell contact for 6 days in vitro.

### 2.9. CD4^+^ T Cell Proliferation Assay

To analyze their capacity to decrease T cell proliferation, we cocultured macrophages (previously cocultured with MSCs in the experimental conditions for the generation of T-regs, Section 2.10) with CD4^+^ T lymphocytes isolated by CD4 MicroBeads (Miltenyi Biotec) in a 1:1 ratio (macrophages:T lymphocytes) without cellular contact using a 0.4-μm Transwell system (Corning), for 6 days. The cocultures were grown in the presence of inducing medium M0, M1 or M2. T cells were labeled with carboxyfluorescein succinimidyl ester (CFSE, Thermo Fisher Scientific, Waltham, MA, USA) 2.5 μM and cultured in RPMI medium (HyClone) containing 10% FBS (Biowest) activated with anti-CD2/CD3/CD28 beads (Miltenyi Biotec) (1 bead for each T cell). After 4 days of coculture, T lymphocytes were collected, washed and blocked with FBS (Biowest) at 4 °C for 15 min. After blocking, PE-Cy™5 mouse anti-human CD4 (BD Biosciences) was added for 20 min, then were washed. Acquisitions were made on a spectral flow cytometer Aurora (Cytek Biosciences) and analyzed with FlowJo (Ashland). Macrophages cultured in the absence of MSCs and different polarization conditions were used as controls. Activated T cells cultured in the absence of macrophages were used as a control to normalize percentages.

### 2.10. Generation of Regulatory T Lymphocytes

To analyze the induction of regulatory T lymphocytes, we cocultured macrophages treated with MSCs without cellular contact in a 0.4-μm Transwell system (Corning) for 6 days in the presence of inducer medium M0, M1 or M2, with CD4^+^ T lymphocytes selected using CD4 MicroBeads (Miltenyi Biotec) in a 1:1 ratio (macrophages:CD4 T lymphocytes) in RPMI medium (HyClone) containing 10% FBS (Biowest) activated with anti-CD2/CD3/CD28 beads (1 bead for each T lymphocyte) (Miltenyi Biotec). After 5 days of coculture, T lymphocytes were collected, washed and blocked with FBS (Biowest) at 4 °C for 15 min. After blocking, FITC mouse anti-human CD4 (BD Biosciences) and PE mouse anti-human CD25 (BD Biosciences) antibodies were added. Then, the cells were stained following the instructions of the FoxP3 staining buffer set (Invitrogen), and PE-Cyanine7 mouse anti-human FoxP3 (eBioscience) was added. Acquisitions were made on a BD FACSCanto II flow cytometer (BD Biosciences) and analyzed with FlowJo V10 software. CD4^+^ T lymphocytes cultured in the absence of macrophages were used as a control.

### 2.11. Quantification of Soluble Molecules

To determine the concentration of secreted soluble molecules, supernatants were obtained from MSCs, macrophage cocultures and macrophages alone (control) incubated in inducer medium M0, M1 or M2. The supernatants were stored at −70 °C until use. Cytokine identification was performed using LEGENDplex cytometry beads (BioLegend, San Diego, CA, USA). The kit was used following the supplier’s instructions. The samples were analyzed on the same day at a low acquisition rate on a BD FACSCanto II cytometer (BD Biosciences). The data were analyzed with LEGENDplex software (BioLegend).

### 2.12. Statistical Analysis

For the statistical analysis, the Kruskal-Wallis H test was used, followed by the U-Mann-Whitney post hoc test, to determine significant differences; analyses were conducted using IBM SPSS Statistics V21.0. *p* < 0.05 was considered significant.

## 3. Results

### 3.1. Cells Derived from CeCa and NCx Display Characteristic Membrane Markers, Morphology and Differentiation Capacity of MSCs

The MSCs from the three sources analyzed (six samples for each source, *n* = 6) present the characteristic phenotype of this cell population stipulated by the International Society for Cell and Gene Therapy (ISCT) [27]. Similar to previous results [1,9], BM-, NCx- and CeCa-MSCs were positive for the markers CD90, CD105, CD73, CD13, HLA-ABC, CD29 and CD10, and negative for the hematopoietic markers HLA-DR, CD45, CD34 and CD14 and the endothelial marker CD31 (Appendix A). Similarly, they presented fibroblastoid morphology (Appendix A). Adipogenic differentiation capacity was evidenced by the staining of lipid vacuoles with oil red, and we observed that unlike BM-MSCs, NCx-MSCs and CeCa-MSCs did not show cells with the characteristic adipocyte morphology; however, fibroblastoid cells with intracellular lipid spots were observed (Appendix A). When osteogenic differentiation was evaluated by the detection of alkaline phosphatase activity, it was observed that MSCs from the three sources presented positive activity for this enzyme in a similar way (Appendix A). Chondrogenic capacity was similar in MSCs from the three sources, which was evidenced by Alcian blue staining of sulfated proteoglycans expressed by MSCs and presented in the extracellular matrix (Appendix A). 

We did not observe differences in cell proliferation of MSCs at passage 4 (Appendix A) or 5 from the three sources, which was measured after 7 days of culture.

### 3.2. CeCa-MSCs Increase the Expression of M2 Markers and Decrease the Expression of M1 Markers in Macrophages

Previous studies have indicated that BM-derived MSCs have the ability to inhibit the expression of membrane markers characteristic of M1 polarization, favoring the M2 phenotype in in vitro models [20]. Therefore, we analyzed the effect of NCx-MSCs and CeCa-MSCs on a population of CD14^+^ monocytes with regard to the induction of M1 or M2 macrophage populations in cell cocultures. Individual experiments with MSCs derived from BM (*n* = 6), NCx (*n* = 6) and CeCa (*n* = 6) were performed to determine the fold increase (compared to the control) in the intensity of positive expression of membrane molecules characteristic of M1 or M2 polarization in macrophages cocultured in cell contact for 6 days in vitro. Macrophages in the absence of MSCs were used as controls and were set at a 1-fold increase. No difference was observed between the groups with regard to the percentage of positive cells.

In the cocultures with MSCs and without inducing medium (Figure 1A), BM-, NCx- and CeCa-MSCs decreased HLA-DR expression (0.33 ± 0.11, 0.38 ± 0.13 and 0.72 ± 0.13, respectively; *p* < 0.05) and increased CD14 expression (1.33 ± 0.20, 1.34 ± 0.15 and 1.19 ± 0.04, respectively; *p* < 0.05). Interestingly, BM-MSCs and CeCa-MSCs increased CD163 (1.73 ± 0.33 and 1.85 ± 0.48, respectively; *p* < 0.05) and CD206 (2.18 ± 0.78 and 1.78 ± 0.29, respectively; *p* < 0.05) expression, in contrast to that observed for NCx-MSCs (0.91 ± 0.06 and 0.49 ± 0.17, respectively; *p* < 0.05). For Arg1 in the M0 condition, BM-MSCs and CeCa-MSCs, unlike NCx-MSCs, increased their expression compared to the control (1.16 ± 0.12 and 1.11 ± 0.08, respectively; *p* < 0.05). For the markers CD80 and CD86, no differences were found. In the cocultures in the presence of inducer medium M1 (Figure 1B), BM-MSCs, NCx-MSCs and CeCa-MSCs decreased HLA-DR (0.50 ± 0.08, 0.52 ± 0.15, 0.62 ± 0.10, respectively; *p* < 0.05), CD80 (0.68 ± 0.10, 0.67 ± 0.13 and 0.75 ± 0.11, respectively; *p* < 0.05) and CD86 (0.78 ± 0.11, 0.53 ± 0.17, 0.82 ± 0.05, respectively; *p* < 0.05) expression and increased CD14 (1.24 ± 0.11, 1.67 ± 0.19, 1.39 ± 0.09, respectively; *p* < 0.05) expression compared to that observed in the control culture. No differences were found for the markers CD163 and CD206. For Arg1 in the M1 condition, BM-MSCs and CeCa-MSCs, unlike NCx-MSCs, increased their expression compared to the control (1.18 ± 0.10 and 1.04 ± 0.03, respectively; *p* < 0.05). In cocultures in the presence of the inducer medium M2 (Figure 1C), BM-MSCs, NCx-MSCs and CeCa-MSCs decreased HLA-DR (0.74 ± 0.12, 0.45 ± 0.11 and 0.78 ± 0.07, respectively; *p* < 0.05) expression and increased CD14 (1.20 ± 0.07, 1.50 ± 0.14 and 1.31 ± 0.15, respectively; *p* < 0.05) expression. Interestingly, in the cocultures in the presence of CeCa-MSCs, an increase was observed in both CD163 (1.65 ± 0.16; *p* < 0.05) and CD206 (1.81 ± 0.14, respectively; *p* < 0.05) expression compared with that in NCx-MSCs (0.27 ± 0.09; *p* < 0.05). CeCa-MSCs significantly increased Arg1 expression (1.05 ± 0.01; *p* < 0.05) in macrophages compared to NCx-MSCs. These results suggest that CeCa-MSCs, in contrast to NCx-MSCs, increased M2 marker expression and decreased M1 marker expression.

### 3.3. CeCa-MSCs Increase the Percentage of Macrophages with Phagocytic Capacity

The increase in the percentage of macrophages with phagocytic capacity is related to their polarization toward an anti-inflammatory (M2) phenotype and that it can be induced by BM-MSCs [28]. With this background, we analyzed the percentage of macrophages with phagocytic capacity cultured in the presence of NCx-MSCs or CeCa-MSCs with or without polarization-inducing media (M1 or M2). We found that BM-MSCs and CeCa-MSCs in the absence of inducing medium increased the percentage of phagocytic cells (90.05% ± 5.38 and 90.98% ± 2.15, respectively; *p* < 0.05) compared with that observed for individual macrophages and in cocultures with NCx-MSCs (79.95% ± 4.22 and 74.50% ± 3.49, respectively) (Figure 2A,B). In cocultures with inducer medium M1, a decrease in the percentage of phagocytic cells was observed in all culture conditions compared with that observed with individual macrophages; however, in the presence of BM-MSCs and CeCa-MSCs, the percentage (73.86% ± 4.90 and 72.33% ± 2.05, respectively; *p* < 0.05) was higher than that observed in cocultures with NCx-MSCs (59.91% ± 4.01) and the control (66.41% ± 4.15) (Figure 2A,B). In cocultures with M2, we observed an increase in the percentage of phagocytic cells in the presence of BM-MSCs and CeCa-MSCs (97.66% ± 0.81 and 96.98% ± 0.50, respectively; *p* < 0.05) compared with that observed for the control and the coculture with NCx-MSCs (93.61% ± 2.33 and 91.76% ± 2.02, respectively) (Figure 2A,B). These results indicate that CeCa-MSCs, unlike NCx-MSCs, have a greater potential to increase the percentage of macrophages with phagocytic capacity, characteristic of an M2 phenotype.

### 3.4. CeCa-MSCs Increase the Intracellular Expression of Anti-Inflammatory Cytokines in Macrophages

M2 macrophages exhibit the decreased expression of proinflammatory molecules such as IFNγ [18] and in turn express anti-inflammatory cytokines such as IDO [29] and IL-10 [18], which regulate the immune response of cells such as T lymphocytes. Placental-derived MSCs increase the intracellular expression of molecules such as IDO and IL-10 in macrophages [19], and BM-MSCs decrease the expression of IFNγ [30] in these cells. To corroborate the M2 phenotype of the macrophages observed in our cocultures, we evaluated their capacity to express IFNγ, IL-10 and IDO. MFI detected in macrophages in the absence of MSCs was considered the control.

In the absence of inducing medium (Figure 3A), the MSCs from the three sources decreased the expression of IFNγ in macrophages compared with that observed for the M0 control. As expected, M1 increased this expression (1.22 ± 0.09-fold increase, *p* < 0.05), and M2 decreased it (0.85 ± 0.05-fold increase, *p* < 0.05). Unlike NCx-MSCs, BM-MSCs and CeCa-MSCs increased the expression of IL-10 (1.26 ± 0.14 and 1.15 ± 0.02-fold increase, respectively; *p* < 0.05) and IDO (1.35 ± 0.13 and 1.18 ± 0.06-fold increase, respectively; *p* < 0.05) in macrophages in coculture (Figure 3A). In the presence of inducer medium M1, NCx-MSCs increased the expression of IFNγ in macrophages (1.37 ± 0.14-fold increase; *p* < 0.05) compared with that observed in BM-MSCs and CeCa-MSCs (0.99 ± 0.11 and 0.97 ± 0.12-fold increase, respectively; *p* < 0.05). With BM-MSCs, NCx-MSCs and CeCa-MSCs, we did not observe differences in the increase in IL-10 (1.31 ± 0.11, 1.41 ± 0.11 and 1.56 ± 0.36-fold increase, respectively; *p* < 0.05) and IDO (1.32 ± 0.15, 1.19 ± 0.17 and 1.16 ± 0.10-fold increase, respectively; *p* < 0.05) expression in macrophages, but they did increase with respect to the control (Figure 3B). In the presence of inducer medium M2, the three sources of MSCs did not modify the expression of IFNγ compared with that observed for the M2 control. However, unlike NCx-MSCs, BM-MSCs and CeCa-MSCs increased the expression of IL-10 (1.24 ± 0.07 and 1.17 ± 0.05-fold increases, respectively; *p* < 0.05) and IDO (1.19 ± 0.08 and 1.12 ± 0.10-fold increases, respectively; *p* < 0.05) in macrophages in coculture (Figure 3C). These results indicate that CeCa-MSCs have a greater potential to increase the intracellular expression of IL-10 and IDO than NCx-MSCs.

### 3.5. CeCa-MSCs Increase the Capacity of Macrophages to Decrease the Proliferation of CD4^+^ T Cells

It has been described that macrophages with the M2 phenotype have the capacity to decrease T lymphocyte proliferation due to the participation of immunosuppressive molecules such as PD-L2 [31]. Moreover, macrophages primed with BM-MSCs without cell contact decrease the proliferation of activated CD4^+^ T cells [21]. We evaluated the ability of macrophages with M2 phenotype generated in coculture with CeCa-MSCs to decrease the proliferation of activated CD4^+^ T cells.

In the absence of inducing medium, macrophages generated in the presence of BM-MSCs and CeCa-MSCs decreased the percentage of proliferating CD4^+^ T cells (73.31% ± 15.72 and 74.02% ± 21.21, respectively; *p* < 0.05) compared to the controls (Figure 4A,B). Interestingly, macrophages in coculture with CeCa-MSCs and in the presence of the M1 and M2 inducing mediums (42.04% ± 26.96 and 59.93.02% ± 29.94, respectively; *p* < 0.05), decreased CD4^+^ T cell proliferation compared to cocultures of NCx-MSCs and controls (Figure 4A,B). These results indicate that macrophages from cocultures with CeCa-MSCs increase their capacity to decrease CD4^+^ T cell proliferation, suggesting their polarization towards the M2 phenotype.

### 3.6. CeCa-MSCs Increase the Ability of Macrophages to Induce the Generation of T-Cell Subsets Displaying a Regulatory Phenotype

M2 macrophages have the ability to generate regulatory T lymphocyte populations [18]. Previous studies have indicated that BM-MSCs generate macrophages with an increased capacity to promote the regulatory phenotype (CD4^+^CD25^+^FoxP3^+^) of T lymphocytes [21]. Thus, we evaluated whether the macrophages obtained in the cocultures with CeCa-MSCs had the ability to favor the generation of Tregs, a process that could be related to the intracellular expression of IL-10 in this macrophage population because IL-10 has been shown to favor the generation of Tregs [32].

In the absence of inducing medium, macrophages generated in the presence of BM-MSCs and CeCa-MSCs increased the percentage of Treg lymphocytes with the CD4^+^CD25^+^FoxP3^+^ phenotype (15.65% ± 2.49 and 15.91% ± 1.25, respectively; *p* < 0.05) comparison with that observed for macrophages in the presence of NCx-MSCs (8.51% ± 1.35; *p* < 0.05). In the presence of inducer medium M1 or M2, BM-MSCs and CeCa-MSCs increased the generation of Treg lymphocytes (M1: 15.76% ± 1.45 and 19.21% ± 1.84, respectively; *p* < 0.05; M2: 23.38% ± 4.20 and 21.35% ± 3.84, respectively; *p* < 0.05) compared with that observed for NCx-MSCs (M1: 12.69% ± 2.04 and M2: 17.5% ± 4.49; *p* < 0.05) (Figure 5A–B). These results indicate that unlike NCx-MSCs, CeCa-MSCs induce the polarization of M2 macrophages with a greater potential to favor the generation of CD4^+^CD25^+^FoxP3^+^ Tregs.

### 3.7. Secretion of Anti-Inflammatory Molecules in CeCa-MSC Cocultures

The presence of BM-MSCs in coculture with macrophages decreases the concentration of soluble proinflammatory molecules and increases anti-inflammatory molecules because of the ability of these cells to favor the polarization of M2 macrophages [20,33]. Due to the above, the concentrations of soluble inflammatory and anti-inflammatory molecules in supernatants were evaluated in our cocultures with MSCs to determine if there was a correlation with the presence of M2 macrophages.

In the absence and presence of inducing medium, in the cocultures with CeCa-MSCs, the concentration of IL-10 increased (M0: 56.49 pg/mL ± 29.70; *p* < 0.05; M1: 356.82 pg/mL ± 42.17; *p* < 0.05; M2: 60.98 pg/mL ± 29.84; *p* < 0.05) with respect to that observed for NCx-MSCs and the control (M0: 9.94 pg/mL ± 6.53 and 3.58 pg/mL ± 0.90, respectively; *p* < 0.05; M1: 62.02 pg/mL ± 10.15 and 117.5 pg/mL ± 83.28, respectively; *p* < 0.05; M2: 26.18 pg/mL ± 13.69 and 7.61 pg/mL ± 2.79, respectively; *p* < 0.05) (Figure 6A–C). In cocultures with inducing media, the presence of CeCa-MSCs increased the concentration of the interleukin-1 receptor antagonist (IL-1RA) (M1: 276.41 pg/mL ± 147.28; *p* < 0.05; M2: 166.60 pg/mL ± 46.84; *p* < 0.05) compared with that observed for NCx-MSCs or the control (M1: 52.85 pg/mL ± 12.02 and 153.95 pg/mL ± 20.25, respectively; *p* < 0.05; M2: 91.64 pg/mL ± 26.66 and 60.22 pg/mL ± 3.67, respectively; *p* < 0.05) (Figure 6A–C). In contrast, the highest concentrations of two proinflammatory molecules, TNFα and IL-1β, were observed in cocultures of NCx-MSCs in the presence of M1 (767.32 pg/mL ± 383.49; *p* < 0.05 and 322.46 pg/mL ± 55.35; *p* < 0.05, respectively) (Appendix A). These results indicate that the presence of CeCa-MSCs, unlike NCx-MSCs, favors an increase in the concentrations of soluble anti-inflammatory molecules and a decrease in the concentrations of proinflammatory molecules in a coculture system with macrophages, an effect that is related to their ability to polarize macrophages toward an M2 phenotype.

### 3.8. Intracellular Expression of M-CSF and IL-10 in CeCa-MSCs in Coculture

M-CSF and IL-10 are two molecules required for M2 polarization in macrophages [34]. BM-MSCs are capable of secreting M-CSF [35], and in systems with macrophages, they promote a greater presence of IL-10 in the medium [20], favoring M2 polarization. In the present work, the intracellular expression of M-CSF and IL-10 was evaluated in MSCs from three sources in coculture with macrophages.

In the absence or presence of inducing medium, similar percentages of MSCs were positive for intracellular M-CSF expression in the three sources of MSCs (Figure 7A,B). However, interestingly, we observed an increase in the percentage of CeCa-MSCs positive for the intracellular expression of IL-10 (M0: 82.6% ± 10.40; *p* < 0.05; M1: 86.12% ± 10.73; *p* < 0.05; M2: 87.75% ± 13.26; *p* < 0.05) compared with that observed in NCx-MSCs (M0: 53.9% ± 6.01; *p* < 0.05; M1: 58.9% ± 9.95; *p* < 0.05; M2: 55.75% ± 3.01; *p* < 0.05) (Figure 6A,C). These results suggest that the ability to express cytokines such as M-CSF and the increase in the expression of IL-10 in CeCa-MSCs could favor the polarization of macrophages toward the M2 phenotype (Figure 8), an effect that is related to higher concentrations of IL-10 in the supernatant of the cocultures with CeCa-MSCs than in the supernatant of the cocultures with NCx-MSCs (Figure 6).

## 4. Discussion

MSCs were identified for the first time in the bone marrow [36]. Since then, the presence of these cells has been reported in different tissues [1], and it has been shown that they have an inhibitory effect on inflammation in a wide variety of immune cells [10]. BM-MSCs exert an effect on macrophage polarization, favoring the anti-inflammatory or M2 phenotype and inhibiting the proinflammatory or M1 phenotype [20]. It has been reported that a tumor is not only composed of tumor cells but also requires the participation of accessory cells of the microenvironment that, through different mechanisms, favor tumor progression. Among the important cells that make up the cancerous stroma, we found macrophages and MSCs derived from tumors [37]. Previous studies have indicated that MSCs derived from lung adenocarcinoma tumor tissue exert an inhibitory effect on NK cells [13]. Similarly, our research group reported that CeCa-MSCs inhibit the cytotoxic effect of CD8^+^ T lymphocytes on tumor cells [9]. In cervical cancer, the presence of macrophages with the M2 phenotype (CD14^+^CD163^+^PD-L1^+^) may be a factor that decreases patient survival [25].

In the present study, similar to that previously published by our group [9], MSCs derived from CeCa and NCx presented fibroblastoid morphology in in vitro cultures. NCx-MSCs and CeCa-MSCs were found to have a low potential for adipogenic differentiation compared to BM-MSCs; however, the three sources had potential for osteogenic and chondrogenic differentiation evidencing their multipotential capacity. Another important characteristic of MSCs is their proliferation capacity, which can be different depending on the source and tissue condition, as we have previously published in MSCs of skin from healthy donors and patients with psoriasis [38]. In our study we observed that despite the difference in the cervical tissue (healthy vs. tumor), the MSCs obtained presented a similar proliferation capacity.

The MSCs expressed membrane markers similar to BM-MSCs, with a positive expression of CD90, CD105, CD73, CD13, HLA-ABC, CD29 and CD10 and negative expression of the hematopoietic markers HLA-DR, CD45, CD14, and CD34 and the endothelial marker CD31. Unlike our previous work where a second passage was used [9], in this study we observed a lower percentage of positive cells in NCx-MSCs and CeCa-MSCs for CD105 and CD73 markers, using passages 4–5. This result is consistent with previous reports that indicate a decrease in the percentage of positive cells for MSCs surface markers when the number of passages increases [39]. In our study we used cells that express the markers indicated by the ISCT [27], which were described as tumor-derived MSCs. However, fibroblast populations called CAFs have been detected in solid tumors [37] and such cells could express some markers similar to MSCs [40]. Previous reports have shown that in CAFs derived from pancreatic ductal adenocarcinoma less than 10% of the population express MSC markers CD90, CD49a, CD44 and CD73; furthermore, it was also observed that CAFs, unlike MSCs, possess low differentiation capacity towards adipocytes, chondrocytes and osteocytes [41]. In our experiments we considered CD90+ cells as MSCs. In this regard, other groups have indicated that FSP1+/α-SMA+ CAFs derived from multiple myeloma had low expression of CD90 and a proportion of these cells express the endothelial marker CD31 [42], which has been reported negative in MSCs [27] as we observed in our study. CD29 and CD10 are two markers that have been proposed as expressed on CAFs. In our study, MSCs from the three sources homogeneously expressed a high percentage of CD29 positive cells. Previous reports indicate that CAFs from breast cancer [43] and metastatic lymph nodes [44] heterogeneously express CD29. CD10 has been detected in breast cancer-derived CAFs, and interestingly, in the coexpression of GPR77 this population is not able to differentiate towards osteogenic or adipogenic lineage [45], in contrast to MSCs which possess such a capacity. It is important to mention that other markers that have been described to define CAFs such as fibroblast activation protein alpha (FAP), alpha smooth muscle actin (α-SMA), vimentin and fibroblast-specific protein 1 (FSP1) [46], would be appropriate to evaluate in our MSCs populations to determine if they could be markers that differentiate CAFs from MSCs.

Our results indicate that the presence of CeCa-MSCs, as well as BM-MSCs, promotes an M2 phenotype in the macrophage population in the absence of any inducing stimulus; the expression of M1 markers such as HLA-DR decrease, and that of M2 markers (CD14, CD163 and CD206) increase. In a previous study, BM-MSCs increased the expression of CD206 in a microglial cell line [47], an effect due to the participation of M-CSF secreted by BM-MSCs [35]. Gastric tumor-derived MSCs have also been shown to increase the expression of CD163 in macrophages [15]. GM-CSF, LPS and IFNγ are molecules that favor the expression of an M1 phenotype in macrophages [34]; however, BM-MSCs, even in the presence of IFNγ/LPS [20] or LPS/ATP [21], are able to inhibit M1 polarization in macrophages, a finding that is consistent with our results. For example, BM-MSCs inhibited M1 polarization in macrophages by decreasing the expression of HLA-DR, CD80 and CD86 as well as increasing the expression of CD14. Interestingly, this decrease in the expression of markers was also observed in a similar way in the presence of NCx-MSCs and CeCa-MSCs, which may be because in the presence of molecules such as IFNγ or LPS, MSCs activate mechanisms that favor their anti-inflammatory effects. They are capable of increasing the molecules that inhibit the inflammatory polarization of macrophages, for example, CD200R, tumor necrosis factor-inducible gene 6 protein (TSG6) [30] and PGE2 [48]. Arg1 has been described as a marker of M2 macrophages, present in both healthy [18,34] and tumor tissue [49]. We found that BM-MSCs and CeCa-MSCs increase Arg1 expression in macrophages unlike NCx-MSCs, which corroborates their ability to polarize macrophage towards the M2 phenotype. Similar to our results, it has been published that BM-MSCs increase Arg1 expression in macrophages in the presence of LPS/ATP [21,30,33]. However, other groups have published that MSCs from healthy tissue compared to those from tumors generate macrophages with lower Arg1 expression, and this effect may be mediated by IL-10 [14].

We found that in the presence of molecules that favor M2 polarization, CeCa-MSCs presented a greater potential to decrease the expression of HLA-DR in macrophages than did the individual M2 inducer medium, a finding that indicates their capacity for M2 polarization. In fact, unlike what was observed with NCx-MSCs, CeCa-MSCs increased the expression of CD14, CD163 and CD206. Contrary to what we observed for the expression of CD206, a characteristic marker of the M2 phenotype, some groups have reported that BM-MSCs in the presence of IL-4 do not modify the expression of this marker [20,30]. This difference may be due to the interaction of M-CSF and IL-13 in conjunction with IL-4 present in the inducer medium evaluated. CD163 is a characteristic molecule of anti-inflammatory macrophages, and in CeCa, the presence of macrophages with this marker is associated with worse patient survival after chemotherapy [24]. Therefore, we hypothesize that CeCa-MSCs may play an important role in the tumor microenvironment by favoring M2 polarization and therefore an anti-inflammatory environment to decrease the immune response against tumor cells.

Phagocytosis has been described as a functional characteristic possessed by macrophages, but this capacity is increased in those with the M2 phenotype [34,50]. Our results indicate that CeCa-MSCs favor the expression of markers of the M2 phenotype in macrophages, for example, CD14, CD163 and CD206, which are associated with macrophages with greater phagocytic capacity [51]. BM-MSCs secrete IL-4, a molecule that increases lysosomal activity in an M2 polarization model of microglia, favoring their phagocytic capacity [47]. This property of promoting phagocytosis was also observed in a coculture with placental-derived MSCs even in the presence of GM-CSF, a molecule that favors M1 polarization [19]. In our coculture systems, we observed that unlike NCx-MSCs, in the presence of BM-MSCs or CeCa-MSCs, the phagocytosis capacity of macrophages increases, an effect that is related to an M2 phenotype. Similar results have been described for macrophages in the presence of BM-MSCs, which favor an increase in the phagocytosis capacity of macrophages through mitochondrial transfer [28] and the secretion of microvesicles [21].

Macrophages produce anti-inflammatory molecules such as IL-10 and IDO and proinflammatory cytokines such as IFNγ, which favor the polarization of other macrophages [18,29]. To evaluate this in the macrophages obtained in the cocultures, we analyzed the intracellular expression of these cytokines. We observed that in the absence and presence of inducer medium M2, and unlike that observed for NCx-MSCs, CeCa-MSCs decreased the intracellular expression of IFNγ and increased the expression of IL-10 and IDO. These results are consistent with those of other studies that indicate that BM-MSCs in coculture decrease IFNγ expression in the presence of IFNγ/LPS in macrophages derived from the RAW264.7 cell line [30] while promoting IL-10 production [52]. IDO is an enzyme that degrades tryptophan by converting it to kynurenine, which inhibits the inflammatory response of T lymphocytes. Previous reports indicate that the presence of kynurenine in medium is a factor that favors phagocytosis in macrophages [53]. Another group showed that the apoptosis of BM-MSCs promotes greater expression of IDO in macrophages and increases their phagocytic capacity [54].

Our results indicate that CeCa-MSCs increase the intracellular expression of IDO and IL-10 in macrophages. IDO exhibits a protumoral effect by decreasing T lymphocyte proliferation [55]. It has been reported that the administration of MSCs in a murine autoimmune uveoretinitis model increases the number of macrophages and the concentration of IL-10, and this effect was related to a decrease in the presence of CD4^+^IFNγ+ and CD4^+^IL-17+ T cells [22]. We observed that CeCa-MSCs increase the capacity of macrophages to decrease the proliferation of CD4^+^ T cells. In this regard, it has been reported that macrophages with this capacity increase the expression of CD163, CD206 [52] and Arg1 [31], which correlates with our results. Similarly, it has been proposed that molecules such as IL-10 [52] and amphiregulin [21] which are both increased in cocultures of macrophages with BM-MSCs, are responsible for decreasing T cell proliferation.

Tregs are present in the tumor microenvironment, where they favor the development of the neoplastic population [56]. In gynecological cancers, a high concentration of TAMs and Tregs is associated with worse survival, and the secretion of IL-10 by TAMs regulates the activation of Treg lymphocytes [57]. In cervical carcinoma, the presence of CD163^+^ macrophages is related to a greater number of infiltrating CD3^+^CD8-FoxP3^+^ lymphocytes and a poor patient prognosis [58]. We observed that CeCa-MSCs polarize macrophages that define an M2 CD163^+^ population and express IL-10. Thus, we evaluated the capacity of this macrophage population to generate Tregs. Our results indicate that CeCa-MSCs have greater potential than NCx-MSCs to favor the ability of macrophages to generate a higher percentage of regulatory T cells with the CD4^+^CD25^+^FoxP3^+^ phenotype. Previous studies indicate that in vitro cultures with interactions between BM-MSCs/macrophages promote the greater generation of CD4^+^CD25^+^FoxP3^+^ Tregs than do cultures in the absence of macrophages [59], and that macrophages previously cocultured with BM-MSCs and subsequently evaluated as a primed population generate a greater amount of CD4^+^CD25^+^FoxP3^+^ Tregs than do macrophages without cocultivation, a result attributed to the production of amphiregulin by said primed macrophages [21].

The coculture of BM-MSCs with macrophages increases the secretion of anti-inflammatory molecules such as IL-10 and PGE2 in the presence of inducing media that favor M1 polarization [20,48]. Similarly, IL-1RA is a molecule expressed and secreted by anti-inflammatory macrophages that can be detected in the conditioned environment of this cell type [29]. Interestingly, we detected the presence of both molecules in the conditioned media of our cocultures and observed an increase in the concentration of both cytokines for CeCa-MSCs compared with that for NCx-MSCs in the presence of inducer medium M1 or M2. Increases in IL-10 are mediated by molecules such as TSG6 [22], COX2 [21], PGE2 [60] and IDO [59], or by the heterodimer CCL2/CXCL12 [52], which can be secreted by MSCs. Similarly, and consistent with our results, MSCs secrete IL-1RA, which favors the secretion of IL-10 and the expression of CD206 in macrophages [61]. Therefore, the concentration of both cytokines in our cocultures with CeCa-MSCs may be a consequence of the macrophage polarization effect toward the M2 phenotype by said MSCs, an effect that is correlated with the decrease that we observed in the concentration of proinflammatory molecules such as TNFα and IL-1β. Similar to our results, it has been reported that the presence of BM-MSCs decreases TNFα in cocultures even in the presence of LPS [33], and the same effect has been demonstrated with tumor-derived MSCs [15]. This effect is driven by molecules such as IL-10 [48].

BM-MSCs have the ability to secrete M-CSF, and in cocultures with macrophages, the concentration of this cytokine increases in conditioned medium and favors the expression of CD206 in macrophages, thus promoting an M2 phenotype [35]. In our results, we observed similar M-CSF expression in both NCx-MSCs and CeCa-MSCs; however, interestingly, we observed an intracellular increase in IL-10-positive cells among CeCa-MSCs compared to NCx-MSCs. This may be related to its greater potential to favor macrophage polarization toward the M2 phenotype that we observed. Our research group previously reported that in the coculture of CeCa-MSCs with CeCa tumor cells, the concentration of IL-10 in the conditioned medium increases and that IL-10 is responsible for decreasing the cytotoxic effect of CD8+ T lymphocytes on tumor cells, an effect that was not observed with NCx-MSCs [9]. Other groups have shown that IL-10 expression is higher in tumor-derived MSCs than in healthy tissue [14] and that the expression of IL-10 by BM-MSCs does not modify the expression of other inflammatory molecules [59]. The detection of M-CSF and the intracellular increase in IL-10 in CeCa-MSCs suggests that both could be responsible for the greater potential for macrophage polarization toward the M2 phenotype compared with that observed for NCx-MSCs. We are currently conducting experiments in which the effects of both cytokines are blocked, so as to determine their participation in this mechanism.

## 5. Conclusions

This is the first study that shows that CeCa-MSCs have a greater potential to decrease M1 polarization and promote M2 polarization in macrophages and to increase the percentage of phagocytic macrophages. We also show that these M2 macrophages, unlike those generated with NCx-MSCs, have higher intracellular IL-10 and IDO expression and favor the generation of CD4^+^CD25^+^FoxP3^+^ regulatory T-cell subsets and the capacity to decrease T cell proliferation. Additionally, we showed differences in the expression of intracellular cytokines between CeCa-MSCs and NCx-MSCs; these cytokines could participate in promoting M2 macrophage polarization in cocultures. MSCs have been postulated as one of the main cellular components of the tumor microenvironment that favor the progression of tumor cells. In light of our in vitro results, we propose that in CeCa, the MSCs present in the tumor stroma can favor the development of neoplastic cells by favoring the M2 polarization of macrophages, which in turn leads to the generation of Tregs, which can contribute to creating an anti-inflammatory environment, thus suggesting a role in antitumor immunity. Finally, although it is important to determine the in vitro immunoregulatory potential of CeCa-MSCs, it is necessary to evaluate these capacities in animal models to analyze the increase in the macrophage M2 subpopulation in the tumor context. These experiments are being planned for future studies.

## Figures and Tables

**Figure 1 cells-12-01047-f001:**
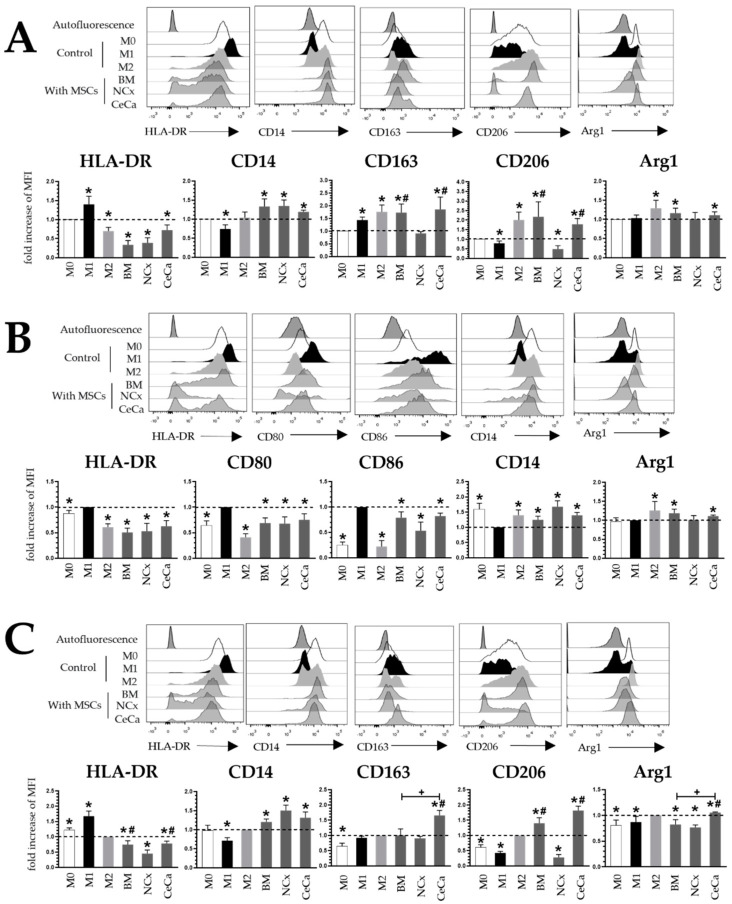
Coculture with CeCa-MSCs increases the expression of M2 membrane markers and decreases M1 markers in macrophages. (**A**) Representative histograms and bar graphs of the mean fold increase with respect to the M0 control for M1 and M2 markers in macrophages cocultured with MSCs in the absence of inducer medium. (**B**) Representative histograms and bar graphs of the mean fold increase with respect to the M1 control for M1 and M2 markers in macrophages cocultured with MSCs in the presence of M1 inducer medium. (**C**) Representative histograms and bar graphs of the mean fold increase with respect to the M2 control for M1 and M2 markers in macrophages cocultured with MSCs in the presence of M2 inducer medium. Bar graphs represent the mean with standard deviation. * Significant difference with respect to control, # significant difference with respect to NCx-MSCs, ^+^ significant difference with respect to BM-MSCs (*p* < 0.05); *n* = 6; dotted line indicates the control for the comparison of fold increase. M0, absence of inducer medium; M1, inducer medium for M1 polarization; M2, inducer medium for M2 polarization.

**Figure 2 cells-12-01047-f002:**
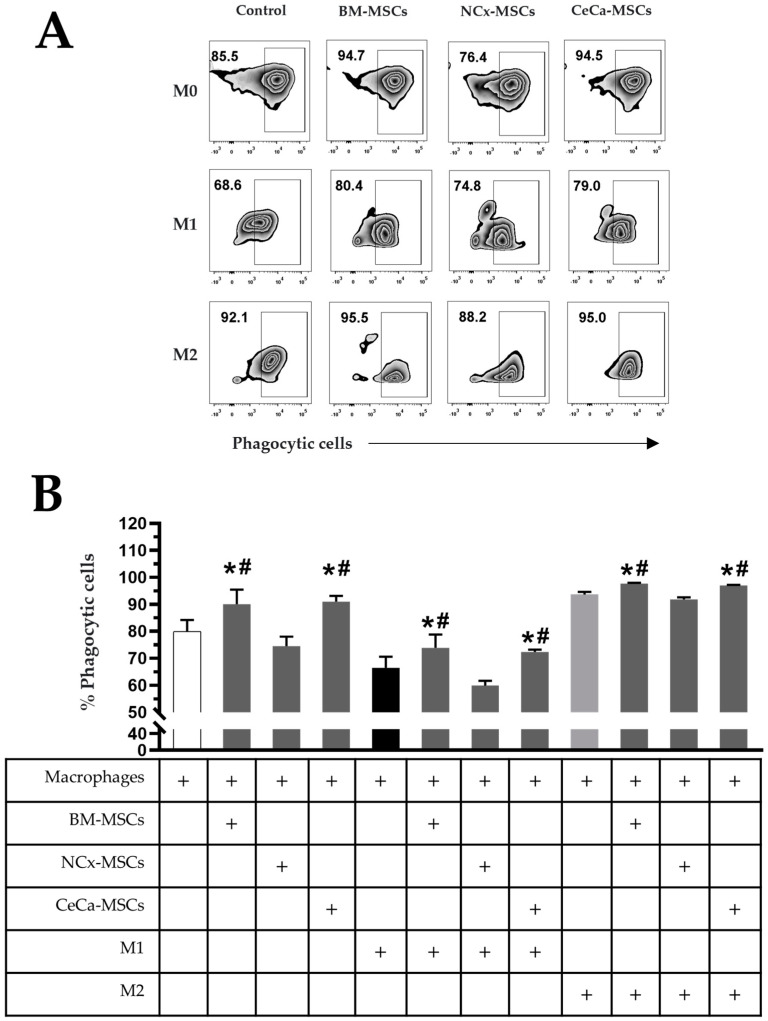
CeCa-MSCs, unlike NCx-MSCs, increase the phagocytosis capacity of macrophages. (**A**) Plot representative of the percentage of phagocytic cells. (**B**) Bar graph of the mean percentage of phagocytic cells evaluated with *E. coli* bioparticles. Bar graphs represent the mean and standard deviation. * Significant difference with respect to the control, # significant difference with respect to NCx-MSCs (*p* < 0.05); *n* = 6. M0, absence of inducer medium; M1, inducer medium for M1 polarization; M2, inducer medium for M2 polarization.

**Figure 3 cells-12-01047-f003:**
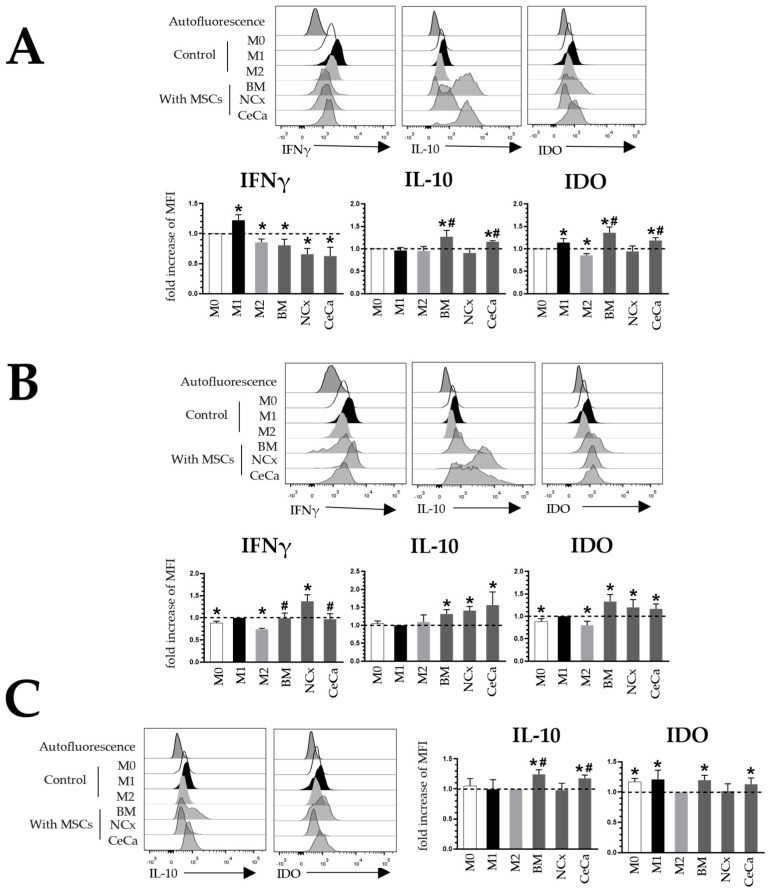
Coculture with CeCa-MSCs and BM-MSCs increases the intracellular expression of IL-10 and IDO in macrophages. (**A**) Representative histograms and bar graphs of the fold increase with respect to the M0 control for intracellular molecules in macrophages cocultured with MSCs in the absence of inducer medium. (**B**) Representative histograms and bar graphs of the fold increase with respect to the M1 control for intracellular molecules in macrophages cocultured with MSCs in the presence of M1 inducer medium. (**C**) Representative histograms and bar graphs of the fold increase with respect to the M2 control for intracellular molecules in macrophages cocultured with MSCs in the presence of M2 inducer medium. Bar graphs represent the mean and standard deviation. * Significant difference with respect to the control, # significant difference with respect to NCx-MSCs (*p* < 0.05); *n* = 6. M0, absence of inducer medium; M1, inducer medium for M1 polarization; M2, inducer medium for M2 polarization.

**Figure 4 cells-12-01047-f004:**
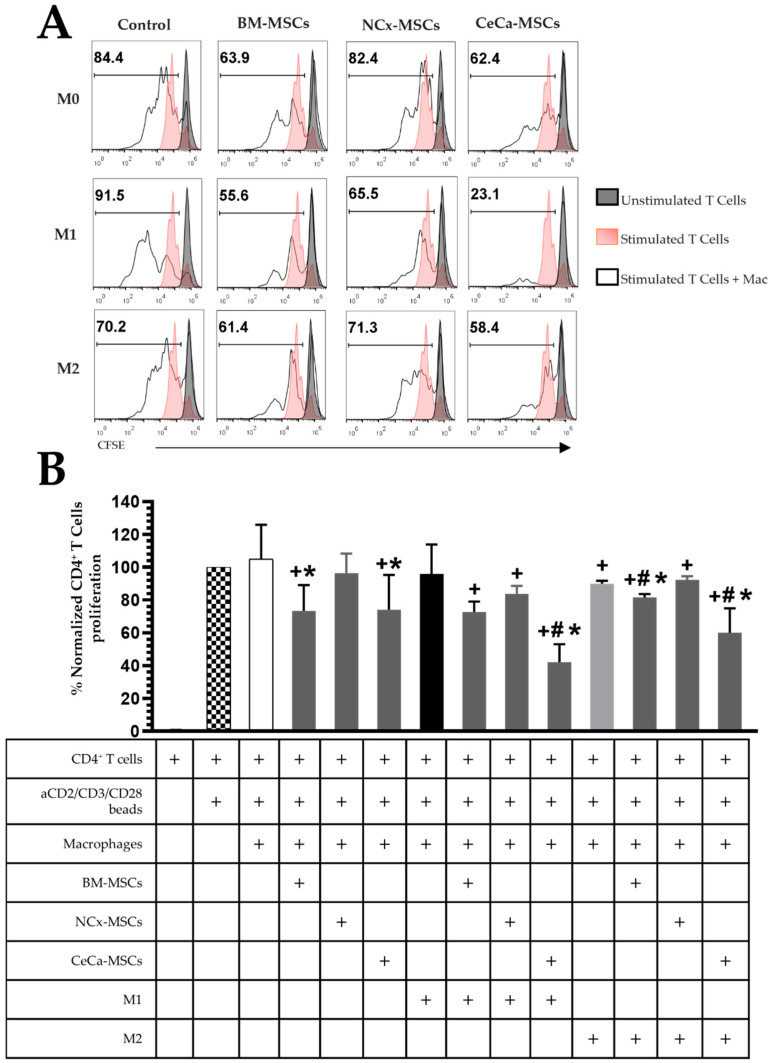
CeCa-MSCs increase the capacity of macrophages to decrease CD4^+^ T cell proliferation. (**A**) Representative histogram of the percentage of CFSE-labeled CD4^+^ T cell proliferation. (**B**) Bar graph of the normalized percentage of proliferative CD4^+^ T cells. Bar graphs represent the mean and standard deviation of the percentage of proliferating CD4^+^ T cells from each condition indicated in the table. Activated CD4^+^ T cells without macrophages were used to normalize the percentage. * Significant difference from the control of macrophages, # significant difference from NCx-MSCs, + significant difference from the control of activated CD4^+^ T cells, (*p* < 0.05); *n* = 6. M0, absence of inducer medium; M1, inducer medium for M1 polarization; M2, inducer medium for M2 polarization.

**Figure 5 cells-12-01047-f005:**
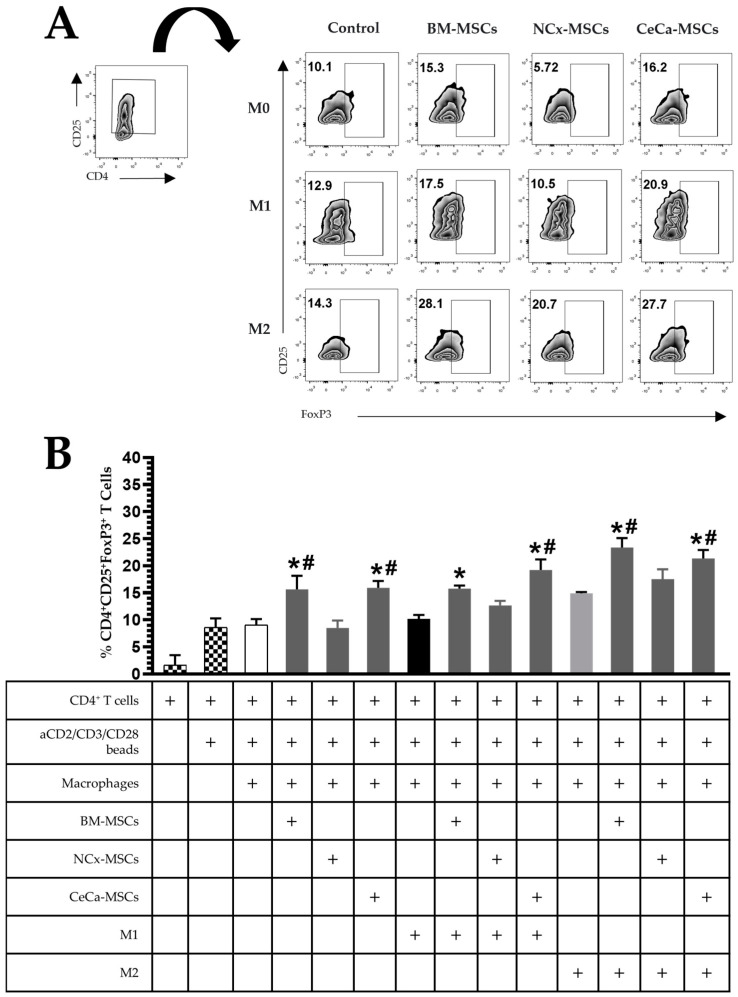
CeCa-MSCs, unlike NCx-MSCs, increase the ability of macrophages to generate CD4^+^CD25^+^FoxP3^+^ regulatory T lymphocytes. (**A**) Plot representative of the percentage of CD4^+^CD25^+^FoxP3^+^ regulatory T lymphocytes. (**B**) Bar graph of the percentage of CD4^+^CD25^+^FoxP3^+^ regulatory T lymphocytes. Bar graphs represent the mean and standard deviation. * Significant difference from the control, # significant difference from NCx-MSCs (*p* < 0.05); *n* = 6. M0, absence of inducer medium; M1, inducer medium for M1 polarization; M2, inducer medium for M2 polarization.

**Figure 6 cells-12-01047-f006:**
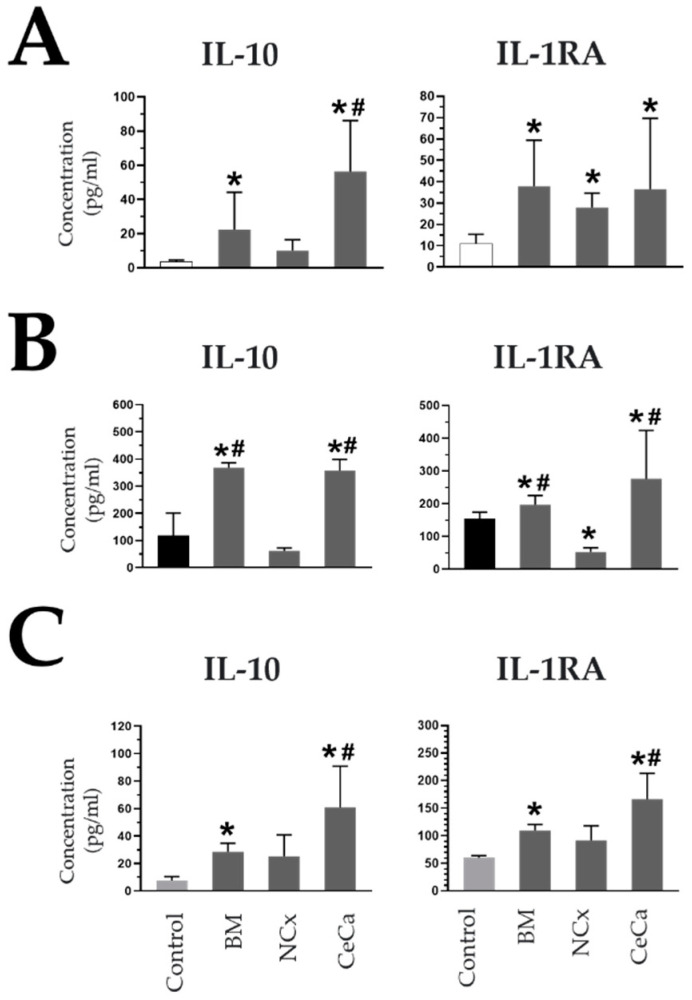
CeCa-MSCs, unlike NCx-MSCs, in coculture with macrophages increase the concentration of soluble anti-inflammatory molecules. (**A**) Evaluation of the concentration (pg/mL) of soluble molecules in cocultures of MSCs with macrophages in the absence of inducer medium (M0). (**B**) Evaluation of the concentration (pg/mL) of soluble molecules in cocultures of MSCs with macrophages in the presence of M1 inducer medium (M1). (**C**) Evaluation of the concentration (pg/mL) of soluble molecules in cocultures of MSCs with macrophages in the presence of M2 inducer medium (M2). Bar graphs represent the mean and standard deviation. * Significant difference from the control, # significant difference from NCx-MSCs (*p* < 0.05); *n* = 6.

**Figure 7 cells-12-01047-f007:**
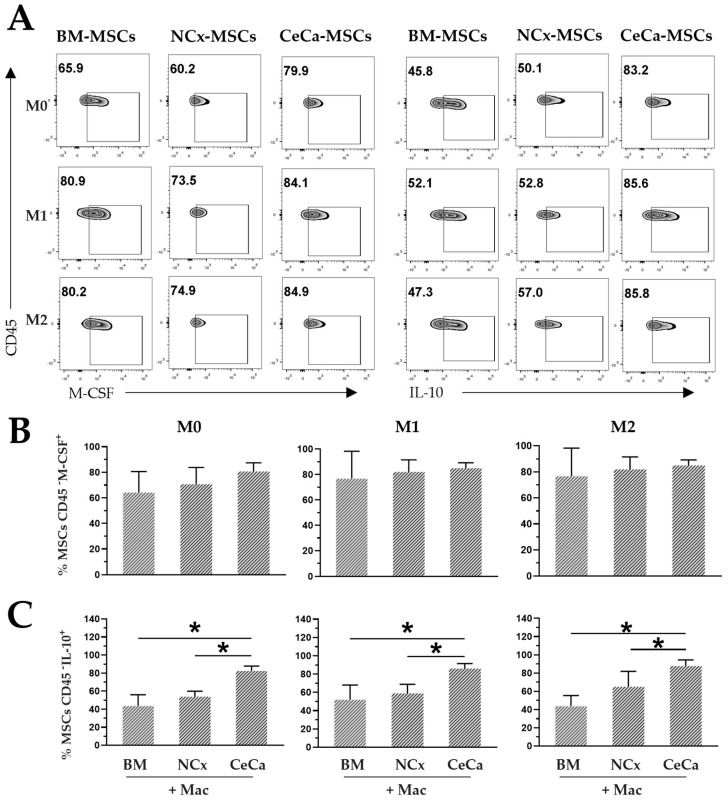
Intracellular expression of M-CSF and IL-10 in CeCa-MSCs cocultured with macrophages. (**A**) Plots representative of the intracellular expression of M-CSF and IL-10 in MSCs from different sources cocultured in the presence of macrophages and M0, M1 or M2 inducing media. (**B**) Percentage of CD45- M-CSF+ MSCs in coculture with macrophages. (**C**) Percentage of CD45- IL-10+ MSCs in coculture with macrophages. Bar graphs represent the mean and standard deviation. * Significant difference (*p* < 0.05); *n* = 6. Mac: macrophages. M0, absence of inducer medium; M1, inducer medium for M1 polarization; M2, inducer medium for M2 polarization.

**Figure 8 cells-12-01047-f008:**
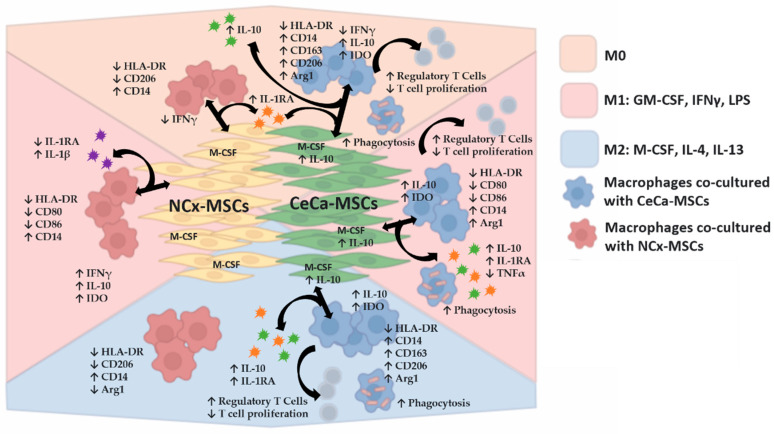
CeCa-MSCs promote the in vitro M2 polarization of macrophages in the presence of molecules that favor an anti-inflammatory or pro-inflammatory phenotype. Schematic diagram of the in vitro mechanism of macrophage polarization toward the M2 phonotype mediated by CeCa-derived MSCs compared with the mechanism of macrophage polarization mediated by NCx-derived MSCs, in terms of the modification of surface markers, intracellular cytokines, phagocytosis in macrophages and also regulatory T cells generation, and decrease in T cell proliferation by these cells. In addition, the diagram shows the expression of intracellular cytokines in MSCs that could participate in such polarization as well as the secretion of cytokines in coculture, in the presence or absence of inducing media that favor M1 or M2 polarization in macrophages. Arrow ↑, increase; arrow ↓, decrease.

## Data Availability

The data presented in this study are available in the article or Appendix A reported.

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
