# Peer review of "Mesenchymal Stem/Stromal Cells Derived from Cervical Cancer Promote M2 Macrophage Polarization"

_cells, 2023, doi:10.3390/cells12071047_

Round 1
Reviewer 1 Report
Comments to the authors
The authors reported that CeCa-MSCs, in contrast to NCx-MSCs, display an increased M2-macrophage polarization potential and suggest a role of CeCa-MSCs in antitumor immunity. However, it is fundamentally necessary to discuss the difference between CeCa-MSCs and cancer-associated fibroblasts (CAFs).
Major points
1. There are many similarities between MSCs and CAFs in terms of cell surface marker expression. Therefore, it is recommended to add more surface markers of CAFs in Supplementary Figure 1.
2. In the author's 2013 article (Stem Cells Dev. 2013 Sep 15;22(18):2508-19), CD73, -90, and -105 of NCx- and CeCa-MSC were observed to be positive in more than 94% of cells, but in Fig S1, the expression of CD73 and CD105 showed a positive rate of 64-78%. Therefore, explanations for these different results should be discussed in the Discussion section. Also, fundamental differences between CeCa-MSCs and cancer-associated fibroblasts (CAFs) need to be discussed.
3. An explanation should be added as to whether the patient completed the tissue donation consent form and whether the institutional research ethics was approved.
4. In the methods part, it was mentioned that six samples of BM, NCx, and CeCa were used, but only three samples were analyzed in some results, which is described in the results section. However, in the figure legend, it was expressed as n=6. It should be indicated accurately the number of samples analyzed in each experiment and explained why only three samples were analyzed.
5. Please check
P-35 culture dishes; 4 °C; FACS CANTO II; Escherichia colli; anti-IFNg-PECy7; -70 °C……
Reviewer 2 Report
Title: Mesenchymal Stem/Stromal Cells Derived from Cervical Cancer Promote M2 Macrophage Polarization
Comments
The present study addressed an actual issue, more precisely investigating how the secretion of mesenchymal stem cells (MSCs) derived from Cervical Cancer influences the polarization of the CD14+ monocytes. The effect of cervical cancer (CeCa-MSCs) cocultures in displaying a high M2-macrophage polarization potential and suggest a role of CeCa-MSCs in antitumor immunity.
The study hypothesis is interesting, addresses an actual research field and is based on relevant bibliographic data. Moreover, the methodology and the experimental procedures are adequate for the study's aims. However, some aspects presented in the paper need clarification, as follows:
- The author analyzed the differentiation capacity of the BM-MSCs, CeCa-MSCs and NCx-MSCs and displayed images. Why did the author not show any result or/ and any significance in differentiation capacity?
- Isolation and culture of NCx and CeCa-derived MSC are not clear in your cited study. The author should provide a protocol for this.
- There is no information regarding the proliferation capacity of the BM-MSCs, CeCa-MSCs and NCx-MSCs. Did the author study the proliferation capacity of the BM-MSCs, CeCa-MSCs and NCx-MSCs? It is also an also essential feature of MSCs.
- The author should add how you calculate M1 and M2 polarization in the Materials and methods section.
Reviewer 3 Report
In this paper the authors show that mesenchymal stem cells derived from cervical cancer promote M2 macrophage polarization which promotes tumor through immune suppression. Overall, the authors have well characterized the macrophage phenotype induced by the MSCs from cancer as compared to normal cervix. The article is well written, logical and interesting. However a few more functional readouts of the M2 macrophage phenotype would strengthen the study. Following are some suggestions
1. Arginase 1 has been strongly corelated with M2 phenotype. How is Arginase1 expression in macrophages cocultured with cancer vs normal MSCs in presence and absence of M2 stimulation?
2. The authors show that M2 polarized macrophages derived from the cocultures with CeCa-MSCs promotes T-regs. M2 macrophages have been shown to render immunosuppression through inhibition of T cell proliferation and causing T cell dysfunction. In the T-reg generation assay, was there any difference in proliferation of the T cells in coculture with CeCa-MSC derived macrophages vs Normal MSC derived macrophages? This data would further clarify the functional aspect of the study.
3. Minor suggestion - The authors may consider to include another assay other than flow cytometry to characterize the M2 macrophage phenotype. For example QPCR or Western blot of the increased expression of some of the M2 markers or decreased expression of any of the M1 markers would strengthen their study.
Round 2
Reviewer 1 Report
Please modify Escherichia colli to Escherichia coli.
Please check the temperature unit. 4oC...